# Characterization, Expression, and Interaction Analyses of *OsMORF* Gene Family in Rice

**DOI:** 10.3390/genes10090694

**Published:** 2019-09-10

**Authors:** Qiang Zhang, Lan Shen, Deyong Ren, Jiang Hu, Guang Chen, Li Zhu, Zhenyu Gao, Guangheng Zhang, Longbiao Guo, Dali Zeng, Qian Qian

**Affiliations:** State Key Laboratory of Rice Biology/China National Rice Research Institute, Chinese Academy of Agricultural Sciences, Hangzhou 310006, China

**Keywords:** OsMORF, gene family, expression analysis, OsMORF–OsMORF interaction

## Abstract

The multiple organellar RNA editing factors (MORF) gene family plays a key role in organelle RNA editing in flowering plants. MORF genes expressions are also affected by abiotic stress. Although seven *OsMORF* genes have been identified in rice, few reports have been published on their expression patterns in different tissues and under abiotic stress, and OsMORF–OsMORF interactions. In this study, we analyzed the gene structure of OsMORF family genes. The MORF family members were divided into six subgroups in different plants based on phylogenetic analysis. Seven *OsMORF* genes were highly expressed in leaves. Six and seven OsMORF genes expressions were affected by cold and salt stresses, respectively. OsMORF–OsMORF interaction analysis indicated that OsMORF1, OsMORF8a, and OsMORF8b could each interact with themselves to form homomers. Moreover, five OsMORF proteins were shown to be able to interact with each other, such as OsMORF8a and OsMORF8b interacting with OsMORF1 and OsMORF2b, respectively, to form heteromers. These results provide information for further study of *OsMORF* gene function.

## 1. Introduction

RNA editing occurs mainly in plastids and mitochondria of flowering plants and plays a key role in post-transcriptional regulation [1]. RNA editing is relatively conserved among flowering plants, with types of base conversion including A-to-I, U-to-C, and C-to-U [2]. Among these conversions, the predominant form in plastids and the mitochondria is C-to-U [2,3]. Previous studies showed that there are 30–40 and 400–500 C-to-U conserved RNA editing sites in chloroplasts and mitochondria, respectively, in flowering plants [1,4]. In plants, the RNA editing sites are recognized by trans-regulatory elements, followed by the catalysis of base conversion [1,3]. The trans-regulatory elements include organelle RNA recognition motif proteins (ORRM), organelle zinc-finger proteins (OZ), protoporphyrinogen oxidase 1 (PPO1), pentatricopeptide repeat proteins (PPR), multiple organellar RNA editing factor (MORF) family proteins also call RNA editing factor interacting protein (RIP) [1,3,5]. Studies have shown that ORRM1 is necessary for plastid RNA editing, whereas ORRM2, ORRM3, ORRM4, and ORRM5 can influence RNA editing in mitochondria [5]. In *Arabidopsis thaliana*, PPO1 and OZ1 were shown to affect 18 and 14 plasmid editing sites, respectively [1,6]. PPR proteins, one of the largest protein families, play an important role in the growth and development of plants, and several PPR proteins have been reported to function in RNA editing [7]. Approximately 200 PSL-type PPR proteins are involved in RNA editing in plastids and mitochondria in *A. thaliana* [7]. In rice, some PPR proteins were identified to regulate the RNA editing of plastids and mitochondria. For instance, OGR1, DUA1, OsPGL1, and PPS1 encode the PPR proteins with a DYW domain in rice [8,9,10,11]. OGR1 was the first PPR–DYW protein identified in rice to affect mitochondrial gene *nad4* editing [8]. DUA1 was shown to participate in the RNA editing of *rpoB* and *rps8* in plastids at low temperature [9]. In addition, OsPGL1, found to be localized in both chloroplasts and mitochondria, is involved in RNA editing of *ndhD* and *ccmFc* in plastids and mitochondria [10]. Moreover, PPS1, located in mitochondria, was shown to be required for RNA editing of *nad3* [11]. WSL5 was also found to encode a P-family PPR protein that influences the RNA editing of *rpl2* and *atpA* at low temperatures [12]. 

The MORF family proteins are small proteins in flowering plants [4]. There are nine, seven, and seven of them in *A. thaliana*, *Zea mays*, and *Oryza sativa*, respectively [10,13]. Research has shown that, in *A. thaliana*, MORF2 and MORF9 are located in plastids; MORF8 is localized in both chloroplasts and mitochondria; and MORF1, MORF3, MORF4, MORF5, MORF6, and MORF7 are targeted to mitochondria [5]. Some MORF proteins can interact with each other by forming complexes, which subsequently influence RNA editing in chloroplasts and mitochondria in *A. thaliana* [5,14]. For example, MORF2 and MORF9 can directly physically interact to form complexes that affect the RNA editing of *ndhD* in chloroplasts, while MORF8 can interact with MORF1 and MORF2 in mitochondria and plastids, respectively [14,15]. In addition, MORFs can also interact with E or DYW–PPR proteins [16]. MORF1 can interact with MEF1 that influences RNA editing in mitochondria in *A. thaliana* [5]. In maize, prediction showed that ZmMORF6 is localized in chloroplasts; however, ZmMORF1 is localized in both mitochondria and chloroplasts, and other ZmMORFs are localized in mitochondria [13]. In rice, *WSP1* (*Os04g0601800*), the homolog of MORF2 in *A. thaliana*, encodes a MORF family protein that is localized in chloroplasts [16]. *WSP1* has been shown to influence the RNA editing of *ndhD*, *rps14*, and *ndhG*, and it can interact with the PPR protein DUA1, which is required for RNA editing of *rps8* in chloroplasts [8,16]. Moreover, in rice, it has been reported that OsMORF2 (Os06g02600), OsMORF8 (Os09g33480), and OsMORF9 (Os08g04450) interact with 10 PPR motifs of OsPGL1 both in vitro and in vivo [10]. 

Previous studies also showed that MORFs might play a key role in the growth, development, and stress resistance of plants. In *A. thaliana*, it was found that *morf2* and *morf9* exhibit reduced chlorophyll content in leaves and *morf8* (*rip1*) exhibits the dwarf phenotype [15,17]. Moreover, in rice, *wsp1* had reduced chlorophyll accumulation in leaves and showed a phenotype of white immature panicles [18]. Furthermore, in poplar, it has been reported that the *PtrMORF* genes respond to drought [3].

In rice, with the exception of *WSP1*, the functions of the *OsMORF* gene family have rarely been reported. Therefore, it is important to further research the biological roles of the MORF gene family. In this study, we analyzed the structure of the MORF genes and then studied the gene expression patterns in different tissues and under various stress conditions. Through yeast two-hybrid (Y2H) experiment, we analyzed all possible relationships among OsMORF proteins and OsMORF–OsMORF complexes in rice. Our results revealed that the characteristics of *OsMORF* genes and provide further information about the functionality of OsMORF.

## 2. Materials and Methods

### 2.1. Sequence Analysis and Phylogenetic Tree of OsMORF Genes

The structures of the OsMORF family genes and proteins were analyzed using the GSD raw method (http://wheat.pw.usda.gov/piece/GSDraw.php). Forty MORF proteins were obtained from *O. sativa*, *A. thaliana*, *Brachypodium distachyon*, *Glycine max*, and *Z. mays* [13]. A phylogenetic tree was constructed via the neighbor-joining (NJ) method implemented in MEGA V7.0 software. For this, a Poisson model was applied, with bootstrapping (1000 replicates) and pairwise deletion for gaps/missing data [13].

### 2.2. Plant Material and Abiotic Stress Treatments

Rice seeds (Nipponbare) were sterilized with 95% ethyl alcohol and washed with sterile water five times. The sterilized seeds were soaked in water for 2 days, and then germinated and transferred to a nutrient solution at 30 °C (16 h light/8 h dark cycle) in a greenhouse. For cold stress, three-leaf-stage seedlings were transferred to an incubator set at 15 °C for 8 days. For salt stress, three-leaf-stage seedlings were transferred to a new nutrient solution containing 10 mM NaCl for 8 days. Leaf samples were collected and stored in liquid nitrogen. At the booting stage, we obtained different tissues of the rice, including roots (R), culm (C), flag leaves (FL), young leaves (YL), mature leaves (ML), and spikelets (S). All experiments involved three biological repeats and each repetition has six rice plants.

### 2.3. RNA Isolation and Quantitative Real-Time Reverse-Transcription PCR (qRT-PCR) Analysis

RNA was extracted using TriZol reagent, in accordance with the manufacturer’s instructions. The first-strand cDNA was acquired using ReverTra Ace qPCR RT Kit (TOYOBO, Japan). The qRT-PCR analysis was performed in accordance with a previous study [19]. The *OsActin1* gene was used as a control, and relative expression levels of genes were calculated by the 2^−ΔΔCT^ method [20]. Related qRT-PCR primers are listed in Appendix A. The significance of the differences between the samples was analyzed using paired Student’s *t*-test, with *p*-values of **p* < 0.05 and ***p* < 0.01 being considered significant.

### 2.4. Yeast Two-Hybrid (Y2H) Analysis

For Y2H analysis, OsMORF-AD and empty BD vector, OsMORF-BD and empty AD vector, and OsMORF-AD and OsMORF-BD vector were co-transformed into the AH109 yeast and grown on synthetic dextrose medium lacking Leu and Trp (SD-T/W) in a 28 °C incubator. Next, the co-transformed yeast strains were transferred to medium lacking Leu, Trp, His, and Ade (SD-T/W/H/A) for further interaction analysis.

## 3. Results

### 3.1. Sequence Analysis of the OsMORF Gene Family in Rice

We obtained seven *OsMORF* sequences from the rice genome, namely, *OsMORF1* (Os11g11020), *OsMORF2a* (Os06g02600), *OsMORF2b* (Os04g51280), *OsMORF3* (Os03g38490), *OsMORF8a* (Os09g04670), *OsMORF8b* (Os09g33480), and *OsMORF9* (Os08g04450). These *OsMORF* genes are distributed in six rice chromosomes: chromosomes 3, 4, 6, 8, 9, and 11. Phylogenetic analysis showed that AtMORF2 and AtMORF8 have two homologous proteins in rice (Figure 1). Similar to the MORFs in *A. thaliana*, these seven OsMORFs were predicted to be located in mitochondria or chloroplasts using TargetP and Wolf PSORT (Figure 1). However, the actual subcellular localization of OsMORFs needs to be verified by experiments. The results of gene sequence-structure analysis showed that the *OsMORF* genes have three to five exons (Figure 2A), the CDSs of *OsMORF* genes range from 498 to 1197 bp (Figure 2A), the proteins encoded by the *OsMORF* genes are 165–398 aa (Figure 2B), and the molecular weights of these proteins are in the range of 24.70–43.31 kDa. In addition, no transmembrane domains in the OsMORF proteins were identified.

In our study, the MORF box was identified in seven OsMORFs, which is similar to the findings in previous studies on maize [13]. Six conserved motifs were identified by comparing the protein sequences of seven OsMORFs (Figure 2B). The results indicated that these OsMORF proteins have motif1, motif2, motif3, motif5, and motif6 (Figure 2B). However, motif4 was found only in OsMORF8a and OsMORF8b (Figure 2B). The MORF box that we identified contains motif1, motif2, motif3, and motif5.

### 3.2. Phylogenetic Comparison of MORFs among Different Species

To investigate the function and molecular evolution of OsMORFs in rice, we constructed an unrooted neighbor-joining (NJ) tree including 40 MORF genes. The sequence information of these MORF genes was obtained from different species, including *O. sativa*, *A. thaliana*, *G. max*, *B. distachyon* and *Z. mays*. According to the NJ tree, the MORF gene family could be divided into six subgroups: subgroups I to VI (Figure 3). Each subgroup contained MOFR genes from different species, with subgroup I being larger than the others, containing 14 MORF genes and accounting for 35% of the total MORFs (Figure 3). The OsMORFs were classified into subgroups I to V, but not subgroup VI (Figure 3).

### 3.3. Expression Pattern Analysis of OsMORF Genes under Cold and Salt Stresses

Many MORF genes play important roles in the growth and development of plants, but few studies have focused on the expression pattern of MORF genes under abiotic stresses, especially in rice [14,17]. Abiotic stresses include cold and salt stresses, which could affect rice growth at the seedling stage; study of the expression pattern of MORF genes under cold and salt stresses in rice is important for clarifying the molecular mechanisms regulating these genes.

Three abiotic stress response gene, *OsOTS1*, *OsNAC6*, and *OsRAN1* were chosen to verify that the salt and cold treatment is suitable. The expression level of *OsOTS1* was decreased in salt and cold stress, while the expression levels of *OsNAC6* and *OsRAN1* were increased under salt and cold stress, respectively (Appendix A). These results were consistent with previous studies [21,22,23], which indicate that stress treatment was properly conducted. To determine whether the *OsMORF* genes were affected by cold and salt stresses, the expression levels of seven *OsMORF* genes were evaluated by qRT-PCR. The results indicated that *OsMORF* expression levels were changed after cold and salt stress treatment. Among the *OsMORFs*, compared with the normal conditions, the gene expression level of *OsMORF8a* increased about 1.7-fold under cold treatment for 5 days; however, its expression level returned to normal after treatment for 8 days (Figure 4A). However, the expression levels of the other six OsMORF genes gradually decreased with the continuation under cold treatment (Figure 4A). The gene expression patterns of these six *OsMORFs* remained consistent under cold stress. The expression levels of *OsMORF2a*, *OsMORF2b*, and *OsMORF9* under salt stress treatment for 5 days showed no significant differences compared with those under normal conditions (Figure 4B). The expression level of *OsMORF8a* was significantly decreased under salt stress treatment for 5 days (Figure 4B). Our studies also showed that the expression levels of all *OsMORF* genes decreased under NaCl treatment for 8 days. These results indicate that the expression pattern of *OsMORF* genes affected by cold and salt stresses, and that the genes of this family may have different sensitivity to cold and salt stresses.

### 3.4. Expression Analysis of MORF Genes in Different Tissues

To confirm the expression patterns of *OsMORF* genes in different tissues in rice, we sampled tissues including roots (R), culm (C), flag leaves (FL), young leaves (YL), mature leaves (ML), and spikelets (S) for qRT-PCR analysis. The experimental results showed that the *OsMORF* genes were expressed in all tissues, but mainly expressed in leaves and spikelets. Among these genes, *OsMORF2a*, *OsMORF2b*, and *OsMORF9* were highly expressed in flag leaves, while *OsMORF1*, *OsMORF3*, *OsMORF8a*, and *OsMORF8b* were highly expressed in mature leaves (Figure 5). Moreover, the expression levels of *OsMORF1*, *OsMORF3*, *OsMORF8a*, and *OsMORF8b* in culm were low (Figure 5). We also found that, among all of the *OsMORF* genes, *OsMORF2b* was most highly expressed in spikes. The expression levels of *OsMORF2b* and *OsMORF9* were lowest in the root (Figure 5).

### 3.5. Analysis of Interactions between MORF Proteins

We obtained the CDSs of seven *OsMORF* genes from the cDNA in rice leaves and cloned them into PGADT7 (AD) and PGBKT7 (BD). The primers used for amplification are shown in Appendix A. The Y2H experiment showed that the AH109 yeast cells that contained BD-OsMORFs and empty AD vector or contained AD-OsMORFs and empty BD vector can grow normally on SD-Leu/Trp medium (SD-L/W), while those yeast cells did not grow on SD-Leu/Trp/His/Ade medium (SD-L/W/H/A) (Appendix A). We found that there were 11 pairs of OsMORF-BD and OsMORF-AD vectors, co-transformed into AH109 yeast cells that can grow on both SD-T/L and SD-T/L/H/A media (Figure 6). The other 17 pairs of OsMORF-BD and OsMORF-AD vectors co-transformed into AH109 yeast cells could only grow on SD-T/L medium (Figure 6). Our results indicate that OsMORF1, OsMORF8a, and OsMORF8b could interact with themselves to form homomers, but this was not the case for the other OsMORFs, including OsMORF2a, OsMORF2b, OsMORF3, and OsMORF9. We also found that OsMORF1, OsMORF2b, and OsMORF3 could interact with OsMORF8a and OsMORF8b, respectively to form heteromers. The results also showed that OsMORF1 could interact with OsMORF3 to form a heteromer. There were two proteins, OsMORF2a and OsMORF9, which interacted with none of the other MORF proteins. These results indicated that five OsMORFs could form homomers or heteromers, with the exception of OsMORF2a and OsMORF9.

## 4. Discussion

RNA editing mainly occurs in plastids and mitochondria, and plays an important role in plant growth and development [5]. The RNA-editosome can recognize nucleotide sequences and edit them, such as substituting C to U via a simple deamination reaction [4]. The RNA-editosome includes PPR proteins, MORF proteins, ORRM proteins, and OZ proteins [4]. These MORF proteins acting as subunits of the RNA-editosome play important roles in RNA editing. In *A. thaliana*, MORF proteins can form homodimers and heterodimers that participate in RNA editing [5,14]. Although the *MORF* genes were reported to respond to abiotic stress in poplar, few reports have been published on the functions of OsMORFs and their responses to abiotic stress in rice [3]. In this study, we provide further insight into the structure of the *OsMORF* genes family, its tissue expression patterns, and responses to cold and salt stresses, as well as the interactions between OsMORF proteins.

Members of the MORF family have been identified in various plants. Reports have described that nine, seven, nine, and seven *MORF* genes have been identified in *A. thaliana*, maize, poplar, and rice, respectively [3,10,13]. Gene structure analysis showed that most of the *OsMORF* genes in rice contain four exons (Figure 2A). Reports have shown that the seven *ZmMORF* (also named *ZmDAL*) genes also have four exons [13]. These results indicate that the *MORF* genes structure is relatively conserved in rice and maize. Moreover, the results predicting the subcellular localization indicated that OsMORF proteins are localized in chloroplasts or mitochondria (Figure 1), which is consistent with the studies of MORF proteins in other plants [3,13]. Similar to the MORF proteins in *A. thaliana* and maize, OsMORF proteins contain small amino acid residues (<500 aa) and have a lower molecular weight [13]. Protein conserved domain analysis showed that OsMORF proteins contained no known domains except the MORF-box domain, which was consistent with the MORF proteins in maize and poplar [3,13]. Phylogenetic analysis indicated that *MORF* genes in various species can be divided into six subgroups (Figure 3). Subgroup I has 13 *MORF* genes, including *OsMORF2b* and *AtMORF2*. In *Arabidopsis*, *morf2* shows an albino seedling phenotype, in which the RNA editing of *ndhD* in chloroplasts is affected [5]. Meanwhile, in rice, *wsp1* (*OsMORF2b*) has reduced chlorophyll content in leaves, along with the RNA editing of *ndhD* in chloroplasts being affected [17]. These results suggest that the function of MORF in the same subgroups might be conserved in different species. Expression analysis showed that seven *OsMORF* genes were expressed in all tissues, and the expression levels of *OsMORF* genes in leaves and spikes were higher than those in the other tissues (Figure 5). This may be due to the leaves and spikes containing numbers of chloroplasts and mitochondria that require *OsMORF* genes to participate in RNA editing.

Previous studies indicated that RNA editing may affect the resistance of plants to abiotic stress [3,8,24]. For example, compared with normal, the decreased or incomplete RNA editing of *RPS14* and *RPS16* was found in soybean under salt stress [24]. It has been reported that the PPR protein that contains the DYW domain DUA1 can interact with WSP1 (OsMORF2b) and form a complex that can respond to cold stress [9]. Therefore, we deduced that OsMORF proteins, the important components of the RNA-editosome, might be involved in abiotic resistance. Our results show that, after cold and salt treatments for 8 days, the expression levels of seven and six *OsMORF* genes are significantly decreased, respectively (Figure 4). Besides, *OsMORF* genes were shown to have different sensitivity to cold and salt stresses. For example, OsMORF8a might have a sensitivity to salt stress rather than cold stress. In addition, after 5 days of treatment with both cold and salt stresses, the expression levels of *OsMORF1*, *OsMORF3*, and *OsMORF8b* were significantly reduced, indicating that these genes were highly sensitive to both stresses. As the expression levels of *OsMORF2a*, *OsMORF2b*, and *OsMORF9* were markedly decreased after 5 days of cold stress rather than salt stress (Figure 4), it is suggested that these three genes were more sensitive to cold stress than salt stress. From the above, under abiotic stress, the reduced expression levels of *OsMORFs* might lead to the reduced efficiency of RNA editing in chloroplasts and mitochondria, thus, affecting the functions of chloroplasts and mitochondria, in turn resulting in the sensitivity of rice to cold and salt stresses.

It has been reported that different *MORF* genes might affect different sets of RNA editing events and some editing events can be affected simultaneously by two or more different MORFs [1]. Some of these MORFs can interact with themselves or with other MORFs, forming homomers or heteromers, respectively, via the protein–protein interaction analyzed [5,14]. In *A. thaliana*, MORF2 and MPRF8 can interact not only with themselves to form MORF2–MORF2 or MORF8–MORF8 homomers, but also with each other to form a MORF2–MORF8 heteromer [14]. In this study, we found that OsMORF1, MORF8a, and MORF8b can interact with themselves to form homomers. While the OsMORF8a and OsMORF8b can interact with OsMORF1, OsMORF2b, and OsMORF3 to form heteromers, respectively (Figure 6). The findings also showed that OsMORF1 can interact with OsMORF3 (Figure 6). These results are consistent with the finding of homologous protein–protein interactions in *A. thaliana*, suggesting that the molecular mechanisms by which these genes regulate RNA editing may be similar in *A. thaliana* and rice. The results also show that OsMORF2a and OsMORF9 did not interact with each other or any other MORF proteins (Figure 6). These two proteins may interact with other proteins such as DYW domain PPR proteins, for example, OGR1, OsPGL1, and so on. It is reported that OsMORF2a and OsMORF9 can interact with OsPGL1, affecting *ndhD* and *ccmFc* editing efficiency [10]. It has been reported that MORF9 interacts with MORF2 in chloroplasts, forming a MORF9–MORF2 complex, which affects the RNA editing of *ndhD* in *Arabidopsis thaliana* [5]. Therefore, the molecular mechanisms by which these two genes regulate RNA editing may differ from those of their homologs *MORF2* and *MORF9* in *A. thaliana*. As such, although the functions of OsMORF and its homolog MORF may be conserved, they have also differentiated among various species. We also noted that there was 65.9% protein sequence similarity between OsMORF2a and OsMORF2b. Compared with the OsMORF2b protein sequence, OsMORF2a lacked the N-terminal (Appendix A). However, OsMORF2a did not interact with either OsMORF8a or OsMORF8b. Therefore, we speculated that the N-terminal of OsMORF2b plays an important role in forming the heteromers.

## 5. Conclusions

In conclusion, this study investigated the MORF family, involved in RNA editing in rice, through gene structure analysis, Y2H experiments, and gene expression analysis. We indicated that OsMORF1, MORF8a, and MORF8b can form homomers with themselves, while OsMORF1, OsMORF3, and OsMORF2b form heteromers with MORF8a and MORF8b, respectively. These homomers or heteromers might participate in the RNA editing in mitochondria or chloroplasts in rice. Besides, under the abiotic stresses of cold and salt, all *OsMORF* genes expression were affected by salt stress, while six *OsMORF* genes expression were affected by cold stress. Our study provides useful information for further studies on *OsMORF* genes regarding the RNA editing mechanism in organelles and their functions in abiotic stress.

## Figures and Tables

**Figure 1 genes-10-00694-f001:**
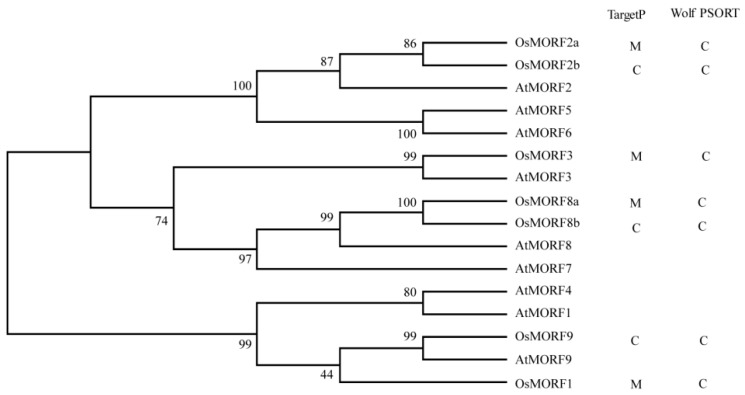
OsMORF family proteins. Multiple sequence alignment of AtMORFs and OsMORFs was implemented using MUSCLE and an NJ tree was built with MEGA v7.0 software. The subcellular localization of OsMORFs was predicted by TargetP (http://www.cbs.dtu.dk/services/TargetP/) and Wolf PSORT (http://wolfpsort.org/). C and M stand for chloroplast and mitochondrion, respectively.

**Figure 2 genes-10-00694-f002:**
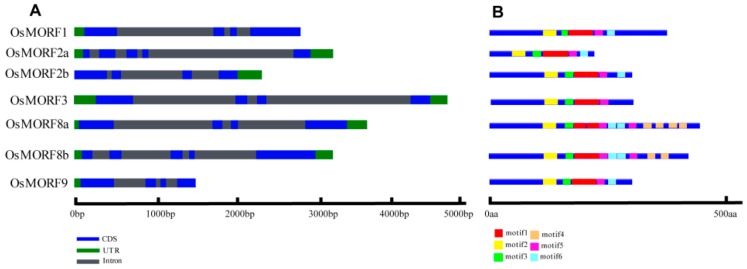
Gene structure (**A**) and protein structure (**B**). The gene and protein structures of OsMORFs were established via GSDraw (http://wheat.pw.usda.gov/piece/GSDraw.php) by submitting both genomic sequences and coding sequences (CDSs) of OsMORF genes.

**Figure 3 genes-10-00694-f003:**
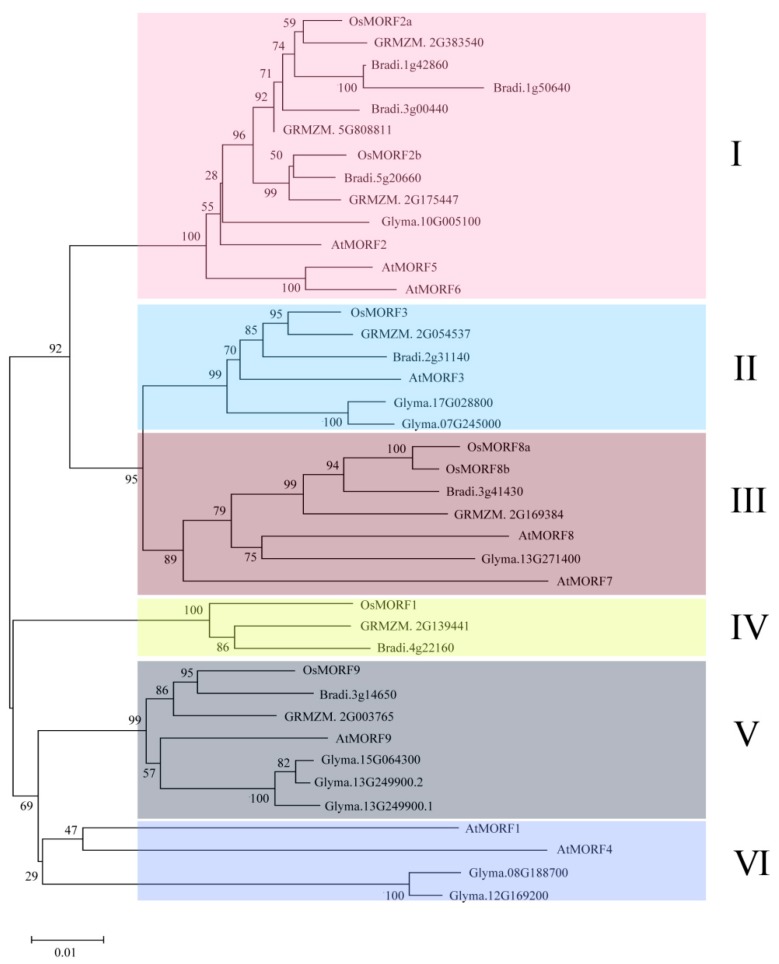
Phylogenetic relationships of the MORF gene family from different plants, including *O. sativa*, *A. thaliana*, *B. distachyon*, *G. max*, and *Z. mays*. The phylogenetic tree was constructed using MEGA v7.0 software.

**Figure 4 genes-10-00694-f004:**
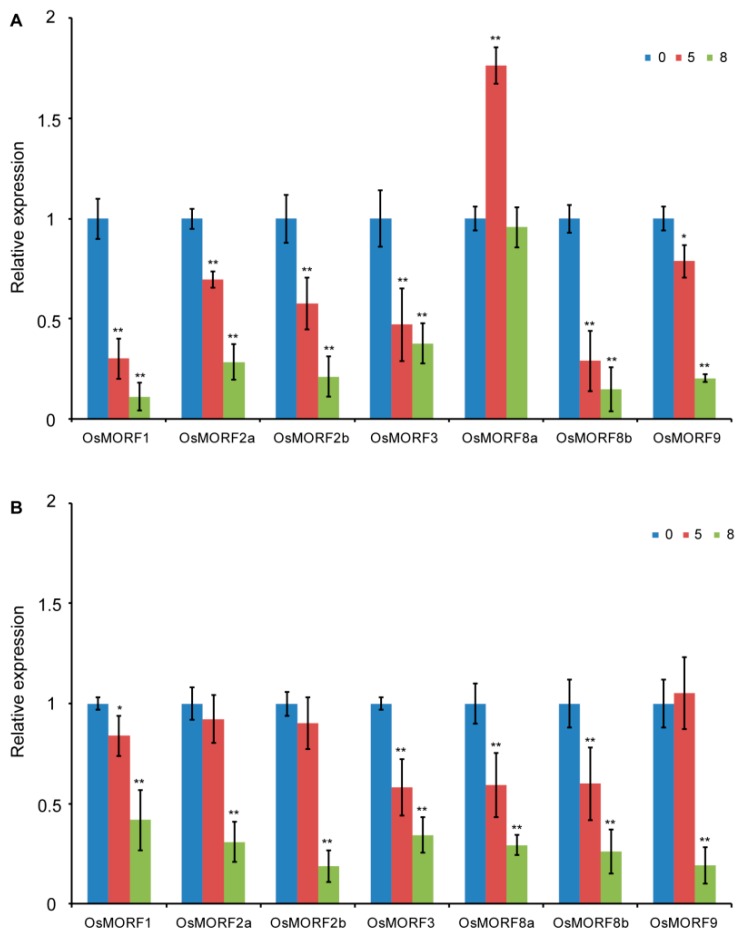
Analysis of *OsMORF* gene expression under cold and salt stress. (**A**) Analysis of *OsMORF* gene expression under cold stress. (**B**) Analysis of *OsMORF* gene expression under salt stress. The data are presented as mean ± SE of three independent replicates. * and ** represent *p* < 0.05 and *p* < 0.01, respectively. 0, 5, and 8 days represent 0, 3, and 8 days of cold treatment, respectively.

**Figure 5 genes-10-00694-f005:**
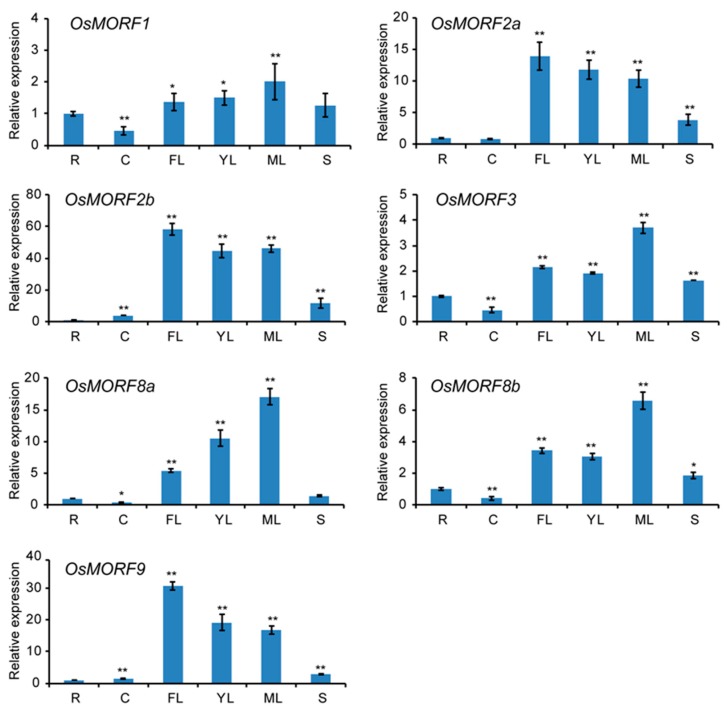
Expression analysis of seven OsMORF genes in various tissues by qRT-PCR. Various tissues were analyzed, including root (R), culm (C), flag leaf (FL), young leaf (ML), mature leaf (ML), and spike (S). The root was defined as 1. The data are presented as mean ± SE of three independent replicates. * and ** represent *p* < 0.05 and *p* < 0.01, respectively.

**Figure 6 genes-10-00694-f006:**
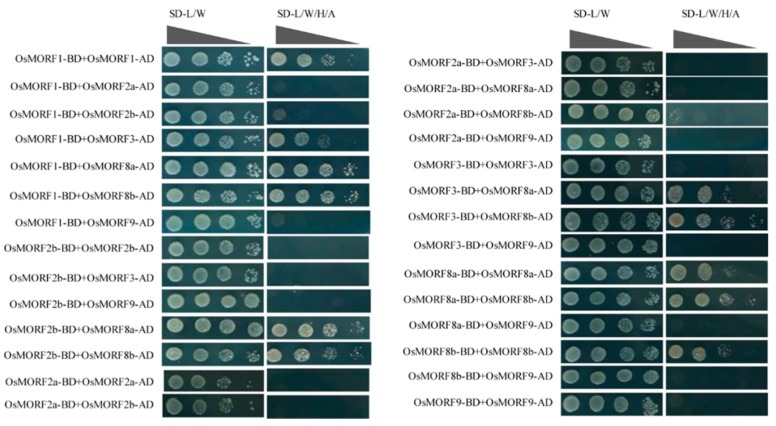
Y2H analysis of all possible OsMORF–OsMORF protein interactions.

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
