# Peer review of "Characterization, Expression, and Interaction Analyses of OsMORF Gene Family in Rice"

_genes, 2019, doi:10.3390/genes10090694_

Round 1

Reviewer 1 Report

The authors conducted gene expression analysis of seven members of rice MORF gene family along with rice developmental stages and also under salt and cold stresses. Furthermore, they analyzed OsMORF-OsMORF interaction using Y2H. The manuscript was written neatly. The scientific results presented in this manuscript will be informative for further study of rice MORF gene function, plant organellar RNA editing, and some abiotic stress research fields.

[Major comment]

In gene expression analysis, the known genes which are upregulated and downregulated under salt and cold stress, respectively, should be included as a reference gene so as to show that stress treatment was properly conducted.

[Minor comments]

The firstly identified mitochondrial RNA editing factor of rice, OGR1 encoding PPR-DYW protein, should be included in the Introduction and Discussion sections. Lines 67-68: Moreover, in rice, it has been reported that OsMORF2 (Os06g02600), OsMORF8 (Os09g33480)….MSU gene ID should NOT be ‘italic’ because here Gene ID means protein. Lines 118-119: (Os11g11020), OsMORF2a (Os06g02600)…. MSU gene ID should be italic. Line 167: were evaluated by quantitative real-time reverse-transcription PCR (qRT-PCR)…, Full name of qRT-PCR should be shown in advance (at Materials and Methods section). Lines 173 & 208: OsMORFs should be italic. Line 231: C to U via a simple deamination reaction[4]. Check spacing between reaction and [4]. Line 254: ndhD in chloroplasts is affected [5](Takenaka et al, 2012). Embedding of OsMORF gene name at each bar graph in Figure 5 would be better for easy recognition by potential readers.

Reviewer 2 Report

This manuscript investigated the classification of the rice MORF genes. Expression profiles of seven OsMORF genes revealed that these genes were responsible to cold and salinity stresses. Protein-protein interactions were assayed using yeast two hybrid system and some of these were shown to be interactive. The data presented seem to be preliminary and further results should be demonstrated.

For functional identification of the OsMORF proteins, please study cellular localization of these proteins. Predictions based on bioinformatics might be useful, however two different algorithms (TargetP and PSORT) showed different results. Also, please mention the meanings to study the interaction between OsMORF2b and OsMORF8a, whose cellular localizations predicted are different and how these can interact in vivo.

Please discuss the contribution of the OsMORF proteins to abiotic stress responses. Because expression of the OsMORF genes was repressed in response to cold and salinity treatments. How repressed expression involves in responses/tolerances to these stresses? I think it’s better to study short-term effects of abiotic stress treatment, for example within 24 hours, because 5 or 8days treatment triggered secondary stress caused by cold and salinity stresses, such as oxidative stress.

Figure 2 legend: Gene structure (A) and protein structure (B)

Figure 4 legend: “0, 5, and 8 days” in the last sentence

Round 2

Reviewer 2 Report

Although I do not have further comments on this manuscript, the author’s claim, “The treatment time of abiotic stress is indeed too long, so our description may be inaccurate.”, looks very odd and if you truly think so, it’s better to redesign the experiment to reach accurate conclusion.